Comprehensive analysis of COMMD10 as a novel prognostic biomarker for gastric cancer

Zhao Wenfang
Lin Jiahui
Cheng Sha
Li Huan
Shu Yufeng
Xu Canxia xucanxia2000@163.com
The Third Xiangya Hospital of Central South University , Changsha , China
Pathania Anup
Electronic publication date: 2023 Mar 9
Publication date: 2023
Volume: 11
Electronic Location ID: e14645
Received 2022 Jul 22; Accepted 2022 Dec 6
Copyright: © 2023 Zhao et al.
Copyright year: 2023
Copyright holder: Zhao et al.
License: This is an open access article distributed under the terms of the Creative Commons Attribution License, which permits unrestricted use, distribution, reproduction and adaptation in any medium and for any purpose provided that it is properly attributed. For attribution, the original author(s), title, publication source (PeerJ) and either DOI or URL of the article must be cited.
License URL: https://creativecommons.org/licenses/by/4.0/

Keywords: COMMD10, Gastric cancer, Prognosis, m6A, Immune infiltration, Biomarker

Funding: National Natural Science Foundation of China 81570509 Changsha Natural Science Foundation Project kq2202118 This work was supported by the following grants and foundations: the National Natural Science Foundation of China (No. 81570509) and the Changsha Natural Science Foundation Project (kq2202118). The funders had no role in study design, data collection and analysis, decision to publish, or preparation of the manuscript.

==============================
Background

COMMD10 has an important role in the development of certain tumors, but its relevance to gastric cancer (GC) is unclear. The purpose of this study is to investigate the difference of COMMD10 expression in gastric adenocarcinoma (STAD) and analyze the correlation between COMMD10 expression and prognosis of STAD patients.

Methods

The expression levels of COMMD10 between STAD and normal tissues were explored using the The Cancer Genome Atlas (TCGA) database. In addition, the expression of COMMD10 in GC was further validated by immunohistochemistry (IHC) staining, qRT-PCR and Western blot. Dot blot experiments were used for exploring m6A expression levels in tissues with high and low COMMD10 expression. Kaplan–Meier analysis and COX regression analysis were used to explore the relationship between COMMD10 and STAD prognosis. A nomogram was constructed to predict the survival probability of STAD patients. GO and KEGG functional enrichment of COMMD10-related genes were performed. The Corrlot software package was used to analyze the correlation between COMMD10 expression levels and m6A modifications in STAD. An analysis of immune infiltration based on the CIBERSOFT and the single-sample GSEA (ssGSEA) method was performed.

Results

COMMD10 expression was significantly associated with multiple cancers, including STAD in TCGA. COMMD10 expression was elevated in STAD cancer tissues compared to paracancerous tissues. COMMD10 upregulation was associated with poorer overall survival (OS), clinical stage, N stage, and primary treatment outcome in STAD. Functional enrichment of COMMD10-related genes was mainly involved in biological processes such as RNA localization, RNA splicing, RNA transport, mRNA surveillance pathways, and spliceosomes. The dot blot experiment showed that m6A levels were higher in cancer tissues with high COMMD10 expression compared with paracancerous tissues. COMMD10 was significantly correlated with most m6A-related genes. COMMD10 was involved in STAD immune cells infiltration, correlated with macrophage cells expression.

Conclusion

High COMMD10 expression was significantly associated with poor prognosis in STAD patients, and its functional realization was related to m6A modification. COMMD10 involved in STAD immune infiltration.

Introduction

COMMD10 is a member of the copper-containing metabolic MURR1 structural domain (COMMD) protein family. The COMMD family is a newly discovered tumor-regulating protein family in recent years. Its family members, COMMD3, COMMD7, and COMMD9 were found to be involved in tumor regulation (Cheng et al., 2022; Zheng et al., 2019; Zhan et al., 2017). In recent years, the relationship between COMMD10 and colorectal cancer and liver cancer has been reported. It has been demonstrated that COMMD10 can inhibit invasion and metastasis of colorectal cancer by targeting the p65 nuclear factor-kappaB (NF-kappaB) subunit and reducing its nuclear translocation, which leads to inactivation of the NF-kappaB pathway (Yang et al., 2017). In addition, COMMD10 can also inhibit hepatocellular carcinoma proliferation and induce apoptosis by blocking the NF-kappaB pathway, and can predict the OS of hepatocellular carcinoma by assessing Barcelona Clinic Liver Cancer (BCLC) staging, providing a basis for the identification of potential therapeutic targets and accurate prediction of prognosis for hepatocellular carcinoma patients (Yang et al., 2021). It has also been shown that in hepatocellular carcinoma, COMMD10 inhibits HIF1α/CP positive feedback loop to enhance radiosensitivity by disrupting Cu-Fe balance. This provides a new target and therapeutic strategy to overcome radioresistance in hepatocellular carcinoma (Yang et al., 2022). These findings indicate that COMMD10 is closely related to digestive system tumors. However, COMMD10 is not currently studied in gastric cancer (GC).

GC is the fifth most common cancer worldwide, and although its incidence and mortality have declined globally over the past 50 years, it remains the third most common cause of cancer death, maintaining a high mortality rate of 75% in most parts of the world, with a median survival rate of less than 12 months in advanced stages. Due to its insidious nature, most patients are not diagnosed until the cancer has reached an advanced stage, increasing poor prognosis and mortality (Smyth et al., 2020; Thrift & El-Serag, 2020; Machlowska et al., 2020). The main treatment for GC remains surgery plus chemotherapy, and although the development of targeted therapies and immunotherapy has improved the prognosis of a proportion of GC patients, however, the available treatment outcomes remain disappointing due to the heterogeneity of the tumor (Joshi & Badgwell, 2021; Li et al., 2021; Zhu et al., 2021). Therefore, identifying effective diagnostic and prognostic factors may help to improve the current treatment strategy for GC patients and thus prolong survival. The present study was designed to investigate the expression of COMMD10 in GC and its prognostic value.

The tumor microenvironment (TME) is the cellular environment in which tumors or tumor stem cells exist and consists of multiple complex components. TME has an impact on tumor growth, proliferation, and migration. Tumor-infiltrating immune cells are an integral component of the TME, and their composition and distribution are thought to be associated with cancer prognosis (Arneth, 2019; Hui & Chen, 2015; Hinshaw & Shevde, 2019). The prognosis of GC is also significantly correlated with the infiltration of different immune cells in its TME (Ma et al., 2022). N6-methyladenosine (m6A) is the most common post-transcriptional modification of mRNA in organisms and can affect the folding, stability, degradation, and cellular interactions of modified RNAs, involving processes such as splicing, translation, export, and decay (Oerum et al., 2021). Accumulating evidence suggests that dysregulation of m6A modification plays an important role in the initiation and progression of a variety of tumors (Sun, Wu & Ming, 2019). In GC, dysregulation of m6A-related gene expression is strongly associated with its progression and prognosis (Guan et al., 2020; Wu et al., 2022). At present, there is no report on the relationship between COMMD10 and tumor immune infiltration and m6A modification. This study is the first to propose the correlation between COMMD10 and immune infiltration and m6A modification of STAD.

In recent years, an increasing number of platforms and databases have enabled researchers to use multiple data sets for bioinformatic analysis of cancer. To better understand the role of the COMMD10 gene in STAD, this study analyzed COMMD10 expression levels in STAD using data available in public databases. The association between COMMD10 expression and clinicopathological features and OS was evaluated. Molecular changes in COMMD10, including genetic alterations, and their impact on survival were explored. COX regression analyses showed that COMMD10 was an independent prognostic factor for STAD. Finally, the relationship between COMMD10 expression and tumor immune infiltration was investigated. Our results provide evidence for a role of COMMD10 in STAD occurrence and prognosis, and may help to identify a potential biomarker for STAD prognosis and treatment.

Materials and Methods

Data download and analysis

mRNA expression data and clinical information were downloaded from the TCGA database (https://cancergenome.nih.gov/) and Genotype-Tissue Expression (GTEx) database. The gene amplification and mutation status of COMMD10 was obtained from the cBioPortal (http://www.cbioportal.org/) for Cancer Genomics. The Human Protein Atlas (HPA) (http://www.proteinatlas.org/) database was obtained for subcellular localization and immunofluorescence images of COMMD10 expression in GC cells.

Survival analysis

The GEPIA database (http://gepia.cancer-pku.cn/) and the TCGA database were used to estimate the correlation between COMMD10 expression and survival in STAD patients with different clinical characteristics. The R package “survival” (version 3.6) was used to obtain an OS survival map for COMMD10. A threshold value of 50% was chosen as the division threshold to divide the cohort into high and low expression groups. Visualization was performed using ggplot2.

Construction and evaluation of prognostic nomogram

All independent clinicopathological prognostic factors from a Cox regression analysis were selected to construct a line plot to assess the 1-, 3-, and 5-year OS probabilities for patients with STAD. The accuracy of the nomogram was validated by comparing the predicted probabilities of the line graph with the observed actual probabilities through calibration curves. The overlapping reference lines indicate that the model is accurate. The RMS R package (https://cran.r-project.org/web/packages/rms/index.html) and survival R package were used to generate a nomogram. C-index was used to determine the discrimination of the nomogram, which was calculated by a bootstrap approach with 1,000 resamples.

Correlation and gene set enrichment analysis

To understand the biological processes and pathways in which COMMD10 may be involved, correlation analyses between COMMD10 and other mRNAs in GC were performed using data collected from TCGA. Genomic ontology (GO) terminology and Kyoto Encyclopedia of Genes and Genomes (KEGG) pathway studies were performed for significantly co-expressed genes using the clusterProfiler package in R language.

Association of COMMD10 expression levels with m6A modifications in STAD

The association of COMMD10 expression levels with m6A-related genes, including YTHDF1, YTHDF2, YTHDF3, YTHDC1, YTHDC2, WTAP, RBM15, RBM15B, ZC3H13, HNRNPC, METTL14, METTL3, IGF2BP1, IGF2BP2, IGF2BP3, RBMX, HNRNPA2B1, VIRMA, FTO and ALKBH5 was analyzed utilizing the R statistical computing language. Correlations between genes were analyzed using the corrlot software package.

Cell culture

Human normal gastric epithelial cells (Ges-1) were obtained from Guangdong Hybribio Biotech Co., Ltd (Guangdong, China). And GC cell lines AGS, MNK-45, HGC-27, SGC-7901 were obtained from Hunan Fenghui Biotechnology Co., Ltd (Hunan, China). Ges-1, HGC-27, MNK-45 were cultured in RPMI-1640 medium (Gibco, USA) plus 10% fetal bovine serum (Procell, Wuhan, Hubei, China) and 1% penicillin and streptomycin (ABT920; G-clone, Beijing, China). The basal medium for AGS and SGC-7901 were Ham’s F12 (Procell, Las Vegas, NV, USA) and Dulbecco’s Modified Eagle Medium (DMEM) (Gibco, Waltham, MA, USA), respectively, and the rest of the culture conditions were the same. All were incubated at 37 °C in a humidified incubator with 5% CO2.

RNA extraction, reverse-transcription RNA, and quantitative real-time polymerase chain reaction

Total cellular RNA was extracted using TRIzol reagent (YEASEN, Shanghai, China), according to the instructions. Reverse transcription was performed using HiScript II Q RT SuperMix for qPCR kit (R223-01; Vazyme, Nanjing, Jiangsu, China) and cDNA was synthesized according to the instructions. cDNA was synthesized using LightCycler 480 II system (Roche, Glyceraldehyde 3-phosphate dehydrogenase (GAPDH) was used as an internal reference. COMMD10 and GAPDH primers were synthesized by Tsingke Biotechnology Co., Ltd (Beijing, China). The primer sequences were as follows:

COMMD10-forward: 5′-ATGGCGGTCCCCGCGGCGCT-3′

COMMD10-reverse: 5′-TCATGTAAGGGAATCCAGCTG-3′

GAPDH-forward: 5′-GGTCACCAGGGCTGCTTTA-3′

GAPDH-reverse: 5′-GGATCTCGCTCCTGGAAGATG-3′

Western blot

A total of 10 paired cancer and paracancerous tissues were collected from December 2021 to March 2022 from gastric cancer patients who underwent surgical resection in the Department of Gastrointestinal Surgery at the Third Xiangya Hospital of Central South University, and written informed consent was obtained from all patients prior to the study. The research was approved by Ethics Committee of Xiangya Third Hospital, Central South University (22101). The methods for tissue protein extraction were as follows: tissue clipping, tissue homogenizer homogenization; lysis on ice with RIPA lysis solution; centrifugation; protein quantification using a BCA kit (KGPBCA; KeyGEN BioTECH, Jiangsu, China); appropriate amount of protein up-sampling; electrophoresis; gel cutting; membrane transfer; milk closure; addition of 1:800 dilution of rabbit anti-COMMD10 (123867; ZEN BIO, Chengdu, Sichuan, China); 1:1,000 dilution of rabbit anti-GAPDH monoclonal antibody (GB11002; Servicebio, Wuhan, Hubei, China); incubated overnight at 4 °C; washed the membrane; and added goat anti-rabbit IgG secondary antibody (HRP; Proteintech, Rosemont, IL, USA); incubated at 37 °C for 90 min; membrane washed; developed by ECL luminescence kit (BL520A; Biosharp, Beijing, China); developed and fixed; and the expression level of COMMD10 in cells and tissues was analyzed with GAPDH as the internal reference protein.

Immunohistochemical staining

Ten paired GC cancer and paracancerous tissues were used for IHC staining. Briefly, paraffin sections were dewaxed, antigenically repaired, endogenous peroxidase blocked by hydrogen peroxide, and serum closed, then incubated overnight at 4 °C with 1:100 dilution of anti-COMMD10 primary antibody (123867, ZEN BIO, Chengdu, Sichuan, China). This was followed by the addition of a 1:200 dilution of HRP-labeled goat anti-rabbit secondary antibody (GB23303; Servicebio, Ghent, Belgium). Then the sample was incubated at room temperature for 50 min, and the DAB kit (Servicebio, Ghent, Belgium) developed color. The positive expression of DAB is brownish yellow.

m6A dot blot analysis

Total gastric cancer RNA was extracted and RNA concentration was detected using Nanodrop 2000 (Thermo Fisher Scientific, Waltham, MA, USA). The sample underwent heat denaturation at 65 °C for 10 min in a thermal cycler and was quickly cooled on ice. A total of 1 ug RNA was loaded on nylon membranes. Next, the membrane was cross-linked by UV, blocked with 5% non-fat milk, and incubated with M6A antibody (1:600, HA601049) at 4 °C overnight. Subsequently, the membranes were incubated with HRP-conjugated goat anti-mouse IgG dilution (1:2,000, Proteintech, Rosemont, IL, USA). Finally, the membrane was visualized using a chemiluminescence imaging analysis system. To determine the consistency of loading, membranes were stained with 0.02% methylene blue for 30 min, then rinsed twice with ultrapure water and photographed.

Statistical analysis

Box line plot and scatter plot methods were used to detect the expression level of COMMD10 gene in GC patients. The cut-off value of COMMD10 expression was selected as the median method of gene expression. Spearman correlation analysis was used to explore the degree of correlation between COMMD10-related genes and COMMD10. Univariate and multivariate Cox analyses were used to screen for potential prognostic factors. In all analyses, *, ** and *** indicate p < 0.05, p < 0.01 and p < 0.001, respectively.

Results

COMMD10 expression in GC

Data downloaded from TCGA and GTEx were used to analyze the expression of COMMD10 in pan-cancer. The results revealed significant differences in the expression of COMMD10 in a variety of tumors (Fig. 1A). Based on the role of COMMD10 in GI-related tumors such as liver and colon cancers, we sought to explore whether it also plays an important role in GC. We further evaluated the mRNA expression levels of COMMD10 in cancer and paracancerous tissues of STAD patients, and the results showed that the expression levels of COMMD10 were significantly higher in tumor tissues than in paracancerous tissues (p < 0.01) (Fig. 1B). In addition, the expression of COMMD10 mRNA in normal cell lines and GC cell lines were detected by qRT-PCR. The expression levels in MNK-45 and AGS cells were found to be significantly higher than those in the normal gastric epithelial cell line GSE-1 (Fig. 1C). Western blotting was also used to detect the expression of COMMD10 in 10 pairs of cancer and paracancerous tissues. We observed that COMMD10 protein levels were higher in most cancer tissues than in paired paracancerous tissues (Figs. 1D, 1E). Finally, IHC staining on clinical specimens were performed to further validate the expression level of COMMD10 in GC. The results showed that the expression of COMMD10 was higher in cancer than in paracancerous tissues (Figs. 1F, 1G). In addition, to understand the mutation level of COMMD10 in STAD, we analyzed its genome and copy number. OncoPrint mapping of the COMMD10 gene in STAD patients in the TCGA dataset was analyzed using cBioPortal plots (Fig. 1H), which showed less than 3% gene amplification, missense and deep deletion mutations in COMMD10. These data confirmed that COMMD10 was highly expressed in GC.

Figure 1 Expression levels of COMMD10.

(A) Expression levels of COMMD10 in normal and tumor tissues from TCGA and GTEx databases. (B) Differential expression of COMMD10 between STAD (n = 375) and normal tissues (n = 32). (C) qRT-PCR detection of mRNA expression levels of COMMD10 in Ges-1 and different GC cell lines. (D, E) Western blot detection of the protein expression level of COMMD10 in cancer and paracancer. (F, G) IHC staining of COMMD10 in cancer and paracancerous tissues. Representative images are shown. Score bars, 200, 50 μm. (H) cBioPortal OncoPrint graph showing the distribution of COMMD10 genomic changes in STAD patients. *, ** and *** indicate p < 0.05, p < 0.01 and p < 0.001, respectively.

Correlation between COMMD10 expression levels and clinicopathological characteristics of GC patients

TCGA database was used to analyze the correlation between different clinical characteristics and COMMD10 expression levels in patients with STAD. Specifically, the relationship between COMMD10 expression and ten clinicopathological characteristics of GC patients were investigated, including age, gender, H. pylori infection, T-stage, N-stage, M-stage, pathological stage, residual tumor and OS events (Fig. 2). The results showed that high expression of COMMD10 was significantly associated with T stage (p < 0.05) (Fig. 2D), residual tumor (p < 0.001) (Fig. 2H), and OS events (p < 0.05) (Fig. 2I) in these patients. These data suggested that COMMD10 was significantly upregulated in STAD and correlated with different clinical features.

Figure 2 Expression levels in tumor tissues of patients with different clinical features of COMMD10 in TCGA database.

(A) Gender, (B) age, (C) H. pylori infection, (D) T-stage, (E) N-stage, (F) M-stage, (G) pathological stage, (H) residual tumor, (I) OS events. * and *** indicate p < 0.05 and p < 0.001, respectively.

High COMMD10 affects the prognosis of patients with STAD in different clinicopathological states

To determine whether COMMD10 expression affects patient survival, GC patients were divided into a high COMMD10 expression group (the top 50% of the highest expression samples) and a low COMMD10 expression group (the remaining 50% of samples) using the GEPIA database. Survival analysis was then performed based on the mean COMMD10 expression value. Kaplan–Meier survival analysis showed that high COMMD10 expression was associated with poor prognosis in STAD patients in terms of OS (HR = 1.7, p = 0.00093) and progression-free disease survival (HR = 1.8, p = 0.0025) (Figs. 3A, 3B). Subsequently another Kaplan–Meier analysis was applied based on the TCGA database to analyze the relationship between COMMD10 expression and prognosis in STAD patients. As seen Fig. 3C, patients with high COMMD10 expression had a worse prognosis than those with low COMMD10 expression (HR = 1.56, p = 0.007). However, it showed no significant value in progress free interval (HR = 1.38, p = 0.071) (Fig. 3D). Then subgroup analyses were performed of patients with different clinicopathological status STAD and showed that high COMMD10 expression was significantly associated with poor prognosis in STAD in: patients over 65 years of female patients (HR = 1.85, p = 0.039), T3 (HR = 1.65, p = 0.037), tissue grade G3 (HR = 1.81, p = 0.005), tumor anatomical site: cardia/proximal (HR = 2.46, p = 0.037) and age (HR = 1.73, p = 0.01). These data are shown in (Figs. 3E–3I).

Figure 3 Prognostic relationship between COMMD10 and gastric cancer.

(A, B) Analysis of the relationship between COMMD10 expression and OS and disease progression-free survival of STAD patients based on the GEPIA database. (C, D) Analysis of the relationship between COMMD10 expression and OS and PFI of STAD patients based on the TCGA database. (E–I) Prognostic subgroup analyses of STAD patients with different clinicopathological status.

High COMMD10 expression is an independent risk factor for overall survival

Univariate and multivariate Cox risk regression analyses were used to identify the independent prognostic factors of STAD in terms of age, sex, T stage, N stage, M stage and COMMD10. Univariate and multivariate analyses showed that high COMMD10 expression was an independent prognostic factor for OS in STAD patients (HR = 1.492, 95% CI [1.053–2.113], p = 0.025). In addition, age (HR = 1.792, 95% CI [1.247–2.575], p = 0.002), N stage (HR = 1.782, 95% CI [1.134–2.801], p = 0.012), M stage (HR = 2.181, 95% CI [1.220–3.899], p = 0.009) were also independent prognostic factors (Table 1).

Table 1 Univariate and multivariate Cox regression analyses of clinical characteristics associated with OS of STAD in TCGA.

Characteristics	Total (N)	Univariate analysis	Multivariate analysis	
Hazard ratio (95% CI)	p value	Hazard ratio (95% CI)	P value	
Age	367					
<=65	163	Reference				
>65	204	1.620 [1.154–2.276]	0.005	1.792 [1.247–2.575]	0.002	
Gender	370					
Female	133	Reference				
Male	237	1.267 [0.891-1.804]	0.188			
T stage	362					
T1&T2	96	Reference				
T3&T4	266	1.719 [1.131–2.612]	0.011	1.359 [0.860–2.147]	0.190	
N stage	352					
N0	107	Reference				
N1&N2&N3	245	1.925 [1.264–2.931]	0.002	1.782 [1.134–2.801]	0.012	
M stage	352					
M0	327	Reference				
M1	25	2.254 [1.295–3.924]	0.004	2.181 [1.220–3.899]	0.009	
COMMD10	370					
Low	183	Reference				
High	187	1.565 [1.124–2.179]	0.008	1.492 [1.053–2.113]	0.025	

Construction of prognostic line graphs based on independent prognostic factors

Then these independent prognostic factors including age, N-stage, M-stage, and COMMD10 were used to construct a prognostic nomogram and a calibration curve was plotted to test the validity of the nomogram. The relationship between the four clinicopathological variables (age, N stage, M stage and COMMD10) and the 1-year, 3-year and 5-year survival probabilities of OS was visually displayed (Fig. 4A). As shown in (Fig. 4B), the calibration curves for 1, 3 and 5 years showed the consistency of our results and predicted values, indicating that this COMMD10-based nomogram performed satisfactorily. The c index of 0.645 showed moderate predictive accuracy. In conclusion, this nomogram may be a better model for predicting survival of GC patients compared to individual prognostic factors.

Figure 4 Construction of prognostic line graphs.

(A) Nomogram for predicting OS at 1, 3 and 5 years. (B) Calibration curves for 1, 3 and 5 years.

Functional enrichment of COMMD10-related genes

We perform differential expression analysis between the low and high COMMD10 in STAD using the Limma package. Only protein-coding genes were retained. A total of 900 differential genes, including 204 up-regulated genes and 696 down-regulated genes, were selected according to the |logFC| >1 and p.adj <0.05 screening conditions (Fig. 5A). Next, we analyse the co-expressed genes associated with COMMD10 expression in the TCGA STAD dataset. A total of 2,728 genes were obtained under |cor spearman| >0.3 and p < 0.05, with 2,724 positively associated genes and four negatively associated genes. The top 50 positively correlated genes were shown in the (Fig. 5B). The 2,728 co-expressed genes were studied for GO terms and KEGG pathway using the clusterProfiler package in R language. COMMD10 co-expressed genes were involved in 709 biological processes, 231 cellular components, 157 molecular functions and 40 KEGGs (p.adj < 0.05 and q value < 0.05). The bubble plots showed the top five information of biological processes, cellular components, molecular functions and KEGGs, respectively. GO term annotations indicated that these genes were mainly involved in RNA localization, RNA splicing, ncRNA processing, tRNA metabolic process; ubiquitin ligase complex, chromosomal region, trophoblast region, mitochondrial matrix; ubiquitin-like protein transferase activity, histone binding (Figs. 5C–5E). The KEGG pathway analysis indicated that these genes were mainly involved in signaling pathways such as RNA transport, ubiquitin-mediated protein hydrolysis, autophagy, mRNA surveillance pathway, and spliceosome. (Fig. 5F). Table S1 summarized the GO terms and KEGG pathway details for COMMD10 co-expression enrichment analyses.

Figure 5 Functional clustering and interaction network analyses of COMMD10-related genes.

(A) Volcano map of differential genes. (B) Heat map showing the top 50 genes positively associated with COMMD10 in STAD. (C) Enrichment analyses of BP of COMMD10 co-expressed genes. (D) Enrichment analyses of CC of COMMD10 co-expressed genes. (E) Enrichment analyses of MF of COMMD10 co-expressed genes. (F) Enrichment analyses of COMMD10 co-expressed gene terms in KEGG. *** indicates p < 0.001.

Association of COMMD10 expression levels with m6A modifications in STAD

The functional enrichment of COMMD10-related genes were mainly involved in biological processes such as RNA localization, RNA splicing, RNA transport, mRNA monitoring pathway, and spliceosome. And m6A RNA methylation plays a crucial role in RNA splicing, export, processing, translation and decay. This suggested that the expression level of COMMD10 may correlate with m6A modification. The level of m6A modification in GC tissues with high COMMD10 expression were assessed using dot blot analysis. The results showed that the expression of m6A was higher in GC tissues with high COMMD10 expression compared with adjacent tissues (Fig. 6A). Next the association between COMMD10 expression levels in STAD and 20 m6A-related genes was explored. As shown in Fig. 6B, COMMD10 expression was significantly and positively correlated with YTHDF3, HNRNPC, METTL14, YTHDC2, RBMX, WTAP, METTL3, YTHDF2, FTO, VIRMA, YTHDC1, RBM15, ALKBH5, HNRNPA2B1, YTHDF1, and RBM15B (p < 0.001). Figure 6C showed the scatter plot of COMMD10 expression with 11 m6A-related genes (Spearman r > 0.3, p < 0.001). As shown, COMMD10 expression were significantly positively correlated with YTHDF3 (r = 0.441, p < 0.001), HNRNPC (r = 0.434, p < 0.001), METTL14 (r = 0.431, p < 0.001), YTHDC2 (r = 0.415, p < 0.001), RBMX (r = 0.395, p < 0.001), WTAP (r = 0.377, p < 0.001), METTL3 (r = 0.365, p < 0.001), YTHDF2 (r = 0.360, p < 0.001), FTO (r = 0.355, p < 0.001), VIRMA (r = 0.345, p < 0.001), and YTHDC1 (r = 0.326, p < 0.001). We also observed that YTHDF3, METTL3, FTO, YTHDC1 and YTHDF1 had a poor prognosis in GC (Fig. 6D). This suggests that there may be a regulatory relationship between YTHDF3, METTL3, FTO, YTHDC1 or YTHDF1 and COMMD10, thus affecting the prognosis of patients.

Figure 6 Association of COMMD10 expression with m6A-related genes in STAD.

(A) Dot bolts. (B) TCGA STAD cohort analyzed the association between COMMD10 and the expression of 20 m6A-related genes. (C) Scatter plots were plotted to show the association between COMMD10 and the expression of m6A-related genes including YTHDF3, HNRNPC, METTL14, YTHDC2, RBMX, WTAP, METTL3, YTHDF2, FTO, VIRMA and YTHDC1. (D) Prognostic value of YTHDF3, METTL3, FTO, YTHDC1 and YTHDF1 in gastric cancer. *** indicates p < 0.001.

Correlation of COMMD10 expression with immune characteristics

To explore the correlation between COMMD10 expression levels and tumor immune response, TCGA database was used to investigate immune infiltration in STAD with different COMMD10 expression levels. The results showed that COMMD10 expression in STAD patients was positively correlated with macrophages, dendritic cells (DCs), immature dendritic cells (iDC), eosinophils, central memory T (Tcm) cells, effective memory T (Tem) cells, helper T (Th) cells, Th1 cells, Th2 cells, and negatively correlated with NK CD56 leukocytes (Fig. 7A). Further studies showed that COMMD10 expression was significantly positively correlated with macrophage (r = 0.238, p < 0.001) (Fig. 7B) and iDC infiltration levels (r = 0.197, p < 0.001) (Fig. 7C). This prompted us to explore the relationship between the level of COMMD10 expression and immune infiltration. We found significant differences (p < 0.05) in the levels of infiltration of helper T cells, iDCs and macrophages into immune cells when COMMD10 expression was divided into high and low expression groups (Figs. 7D–7G). It is noteworthy that COMMD10 was closely related to M2 macrophages based on the CIBERSOFT algorithm (Fig. S1).

Figure 7 Correlation analysis of COMMD10 expression and immune infiltration in GC.

(A) The correlation between COMMD10 and immune infiltrating cells in GC. (B) Scatter plot showing the correlation between the expression of COMMD10 and the infiltration level of macrophages. (C) Scatter plot showing the correlation between the expression of COMMD10 and the infiltration level of iDC. (D) Differential distribution of immune cells in patients with high COMMD10 expression and low COMMD10 expression. (E–G) Histograms showing the difference in the infiltration levels of macrophages, iDC and NK CD56 leukocytes between high COMMD10 expression and low COMMD10 expression groups. *, ** and *** indicate p < 0.05, p < 0.01 and p < 0.001, respectively.

Discussion

We found that COMMD10 expression levels were significantly higher in STAD tumor tissues than in paracancerous tissues through the TCGA database, and subsequently we analyzed the differences in mRNA expression levels of different cell lines and protein expression levels of cells and tissues, and found that COMMD10 expression was significantly higher in GC tumor tissues both at the transcriptional and translational levels. It is suggested that COMMD10 may have a pro-carcinogenic effect in GC. Whereas previously in both colorectal and hepatocellular carcinomas, COMMD10 has a cancer suppressive effect (Yang et al., 2017; Yang et al., 2021), which reflects the differences of the role played by COMMD10 in different cancers, and the specific mechanisms need to be further explored.

Valid prognostic biomarkers can provide important information on cancer aggressiveness and/or clinical outcomes of specific untreated patients. Moreover, they are an important component of personalized medicine and precision medicine, as they can prevent under- or over-treatment. Using the TCGA database, we found that COMMD10 expression was positively correlated with tumor T-stage and residual tumor, suggesting that COMMD10 may be involved in the infiltrative growth of STAD. Our subsequent analyses both for OS, progression-free survival of disease, and comparison of survival rates in different subgroups revealed that patients with high COMMD10 expression had lower survival rates, and that high COMMD10 expression was an independent risk factor for OS of STAD patients. Our findings consistently suggest that COMMD10 may serve as a reliable predictor of prognosis in STAD patients.

To further investigate the biological functions of COMMD10 in STAD, we performed functional enrichment of COMMD10-related genes. The results showed that these genes were mainly involved in RNA localization, RNA splicing, ncRNA processing and other processes. m6A RNA methylation plays a crucial role in RNA splicing, export, processing, translation and decay, and is the most common type of mRNA post-transcriptional regulation in organisms. Therefore, we speculate that the expression level of COMMD10 may be correlated with m6A modification. The results of the association between COMMD10 expression levels and 20 m6A-related genes in STAD showed that Commd10 was significantly correlated with most m6A-related genes. The m6A modifications are ubiquitous and common RNA modifications that can affect tumor progression and metastasis by influencing the expression of several cancer-related genes (Chen et al., 2021; Zhao et al., 2021). We also observed that m6A-related genes (YTHDF3, METTL3, FTO, YTHDC1 and YTHDF1) and COMMD10 had a poor prognosis in GC. This suggests that there may be a regulatory relationship between m6A-related genes (YTHDF3, METTL3, FTO, YTHDC1 and YTHDF1) and COMMD10, thus affecting the prognosis of patients. METTL3 is upregulated in GC patients with poor prognosis (Yue et al., 2019), and can promote the progression of GC by targeting the MYC pathway (Yang et al., 2020). In addition, the mRNA of preprotein translocation factor (SEC62) and ARHGAP5 can also be highly modified by aberrant METTL3, leading to accelerated GC progression (He et al., 2019; Zhu et al., 2019). Therefore, we speculated whether COMMD10 could also be highly modified by aberrant METTL3, which could affect the progression of GC.

Given the critical role of the TME in mediating cancer progression and the fact that tumor-infiltrating immune cells are an integral component of the TME, we sought to investigate the relationship between COMMD10 and immune infiltration in STAD. Our study showed that high COMMD10 expression was positively correlated with macrophage and iDC abundance, and negatively correlated with NK CD56 leukocytes abundance. In TME, macrophages, known as tumor-associated macrophages (TAM), are among the most abundant immune cells, and the extent of M2-type TAM infiltration in tumor tissues is positively correlated with poor prognosis in various cancers and can induce immune tolerance during tumorigenesis and progression (Oya, Hayakawa & Koike, 2020; Rihawi et al., 2021; Gambardella et al., 2020), suggesting that COMMD10 promotes macrophage infiltration in the STAD TME and promotes tumor progression. DCs are the most predominant antigen-presenting cells and are an important antitumor component (Fu & Jiang, 2018; Gerhard et al., 2021; Sadeghzadeh et al., 2020), but their antitumorigenicity correlates with the level of DC maturation (Nagarsheth, Wicha & Zou, 2017). It has been shown that tumor growth and angiogenesis are positively correlated with iDCs levels (Fainaru et al., 2010), and high expression of iDCs are associated with worse prognosis in hepatocellular carcinoma (Huo et al., 2021), oral tongue squamous cell carcinoma (OTSCC) (Sales De Sá et al., 2020), and pancreaticobiliary ductal periampullary carcinoma (Lundgren et al., 2017). The iDCs expression were elevated by high expression of COMMD10 in this study, which may suggested a worse prognosis in STAD patients. NK cells have been used in anti-tumor immunotherapy research as an anti-tumor cell (Xie et al., 2020; Cózar et al., 2021; Wu et al., 2020). Previous studies have found that high levels of NK cells infiltration in colorectal cancer (Sconocchia et al., 2014), lung cancer (Jin et al., 2014), esophageal cancer (Xu et al., 2016), and GC (Li et al., 2020) are associated with better prognosis. Thus the results of our study implied that high COMMD10 expression was associated with worse prognosis in GC.

Although our results may provide new insights into the correlation between COMMD10 and STAD, certain limitations were noted in this study. First, there may be sample bias due to data downloaded directly from public databases. Second, to increase the confidence of the results, the sample size should be further expanded. Third, further experimental validation is required to elucidate the biological functions of COMMD10 in vitro and in vivo.

Conclusions

Our study reveals the prognostic value of COMMD 10 in STAD for the first time. Our findings strongly suggest that COMMD10 has potential as a biomarker for predicting treatment outcome and prognosis of STAD patients. However, further experimental validation is required to elucidate the biological impact of COMMD10 on GC and its underlying mechanisms.

Supplemental Information

Supplemental Information 1 Raw data and code.

Click here for additional data file.

Supplemental Information 2 Analysis of immune infiltration based on the CIBERSOFT.

Click here for additional data file.

Supplemental Information 3 GO terms and KEGG pathway details.

Click here for additional data file.

Additional Information and Declarations

Competing Interests

Author Contributions

Field Study Permissions

Data Availability

The authors declare that they have no competing interests.

Wenfang Zhao conceived and designed the experiments, performed the experiments, analyzed the data, prepared figures and/or tables, and approved the final draft.

Jiahui Lin conceived and designed the experiments, performed the experiments, analyzed the data, prepared figures and/or tables, and approved the final draft.

Sha Cheng performed the experiments, analyzed the data, prepared figures and/or tables, and approved the final draft.

Huan Li analyzed the data, authored or reviewed drafts of the article, and approved the final draft.

Yufeng Shu analyzed the data, prepared figures and/or tables, and approved the final draft.

Canxia Xu conceived and designed the experiments, authored or reviewed drafts of the article, and approved the final draft.

The following information was supplied relating to field study approvals (i.e., approving body and any reference numbers):

Field experiments were approved by Ethics Committee of Xiangya Third Hospital, Central South University (project number: 22101).

The following information was supplied regarding data availability:

The raw measurements are available in the Supplemental Files.

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
