# Peer review of "Comprehensive analysis of COMMD10 as a novel prognostic biomarker for gastric cancer"

_PeerJ, doi:10.7717/peerj.14645_

## Round 0.1 · original submission · Major Revisions

The authors are advised to revise as per the reviewer's concerns.

·

Basic reporting

Zhao et al submit an article investigating the role of COMMD10 as a prognostic factor in gastric adenocarcinoma. The authors utilize a plethora of bioinformatics tools, cell-based assays and clinical analysis to demonstrate the prognostic value of this protein to gastric cancer. Nevertheless, there are several aspects of the study that could be drastically improve and increase the overall merit of the manuscript.

On a different note, I would like to congratulate the authors for the great quality of the figure preparation and writing clarity.

Experimental design

To begin with, the experimental design of the study is biased overall. The selection of COMMD10 and gastric cancer seems like a cherry-picking choice and not a product of a sophisticated, unbiased cox-regression model in gastric cancer patients. In case that someone would like to overcome this major bottleneck, we should emphasize on several points of the experimental design that need to be improved.

1. The authors do not outline how the nomogram has been constructed. Given the clinical impact of nomograms in oncology, this information should be clearly stated.
2. The connection between the results and the m6A-RNA modification is still very loose and not clearly explained in the manuscript. The authors should improve this section in order to justify why this analysis was performed. Otherwise, it will be another demonstration of cherry-picking in this manuscript.
3. In similar extent, it is unclear at best, why the authors performed the immune infiltration analysis. Additionally, the most accurate bioinformatics tool for TME infiltration is CIBERSOFT and not ssGSEA. The authors should improve the conceptual connection in their manuscript and perform additional analysis on TME.

Validity of the findings

Regarding the results of the study and the validity of the findings, there are several points that need to be addressed prior to publication of the manuscript :


1. In Figure 1E, the loading of the WB is not even, leading to difficulties in interpreting the changes in levels of COMMD10 between normal and tumor gastric tissues. The authors should provide a Coomassie stain for equal loading or the repeat the WB analysis.
2. In figure 1H, the authors outline the genomic alterations of COMMD10 gene through a Oncoprint. Furthermore, the authors demonstrate that high expression of COMMD10 is associate with poor prognosis in patients with gastric cancer. Nevertheless, based on the data of the oncoprint, the majority of the patients with gastric cancer are presented with deletions of COMMD10 instead of amplification or gain of function. To support their findings, the authors are advised to show KM plots for the patients with deep deletions and amplification in order to support their overall results.
3. In Figure 3, the authors should clarify if the p-value displayed at the KM analysis has been generated through log-rank statistics. If not, all the figures should be replotted using this statistical test.
4. As mentioned above, in Figure 5 and 6 the connection between m6A-RNA methylation, immune infiltration and COMMD10 is weak at best. The authors should include stronger conceptual connection and background in order to make their data solid.

Additional comments

I would like to wish the best of luck to the authors during the revision process.

Reviewer 2 ·

Basic reporting

Though the English writing, literature review and references are sufficiently well quoted, there are minor errors.
The hypothesis is unclear with no premise of why the authors decided to find a relation between COMMD10 and m6A modifications.
It might have an extrapolation from observations in other cancers or It just looks like a serendipitous tumbled association.
Few References are out of formatting order.


Error: The Y-axis in all panels in Figure 2 and others has Log2(TPM+1) for expression, why?

Experimental design

The authors have not provided any experimental evidence for the claim that COMMD10 is associated with Immune infiltration. They should knockout the gene or make mutants of COMMD10 and assess the tumorigenic parameters like migration, invasion, etc. The complete analysis fall short to suffice the title just based on statistical analysis which provide association not direct functional role. Also, the analysis “enrichment of infiltrated macrophages and T-cells” show a poor correlation (should be expected to be nearly or greater than 0.5, to show strong correlation). They should be able to demonstrate by m6A RNA immunoprecipitation in COMMD10 high and low expressing cellines/patient samples; if not RIP at least Dot Blots will also be enough to support their claims. The Fig 5A and other panels does not have METL proteins in neither COMMD10 high or low category with no m6A RNA modiciations being enriched, Could you speculate why?

Validity of the findings

The authors presentation of their experimental evidence is insufficient towards their claim. Supportive experiments using cell lines or patient tumor samples with genetic manipulation are necessary.

Reviewer 3 ·

Basic reporting

In this manuscript, the authors focuses on COMMD10 and investigated the multi-omics profile of gastric adenocarcinoma (STAD) using publically available bioinformatics tools. They also used their own expression data for validation: the expression of COMMD10 was elevated in STAD cancer tissues. The scheme of the study is clear. Basically, I feel that the research is well organized. However, there are several points need to be clarified:

1. It was not clear why the authors focused on COMMD10. According to The Human Protein Atlas (https://www.proteinatlas.org/humanproteome/pathology), 170 genes are listed as potential poor prognostic factors in stomach cancer. Although COMMD10 is included in the list, it was difficult to understand whether focusing on a single COMMD10 gene is sufficient or not, or why the researchers focused on this gene. It would be desirable to have a detailed explanation of this.

Experimental design

no comment

Validity of the findings

2. Figures 1A and 1B. Indeed, differential expression of COMMD10 in STAD is statistically significant, but when I look at other cancer types, some stand out more than others, such as GBM. Why the focus on STAD?
3. Figure 1A. Several normal tissues such as Thyroid (THCA normal group) have higher COMMD10 expression than STAD. What is the function of COMMD10 in normal tissues?
4. Figure 1H. There seems to be a several percentage of STAD patients with deep deletion of COMMD10 gene. Is this correct? What percentage? Is there a correlation between the expression level and amplification/deep deletion?
5. Fig. 5A and l.260: 2,724 genes were positively correlated with COMMD10 and 4 were negatively correlated. Are other genes with high correlations also associated with poor prognosis, as is the case with COMMD10? For example, what about the genes described in l.282-284?

---

## Round 0.2 · Major Revisions

The authors are advised to revise the manuscript as per the reviewers suggestions.

·

Basic reporting

The authors return an improved version of their manuscript, with more clear writing and reduced flaws in the figures and data. I have a couple of more comments regarding methods and data that I will outline in the corresponding section below.

Experimental design

Methods that are still not clear :

1. Still it is not clear for me technically how the authors developed the nomogram. I can see what are the parameters that they used, but not which software and method they used. Since the nomogram are statistically prone to errors, it is important to provide this information.

Validity of the findings

Data that need editing and clarification

1. In figure 5A and 6B the authors mention about the ''correlation'' of COMMD10 with all these genes, providing as well a p-value claim. It is still unclear to me how these statistical analysis has been performed. The authors should perform differential expression analysis between the high and low COMMD10 STAD patients.

Additional comments

The manuscript needs some final additional analysis prior to submission.

Reviewer 2 ·

Basic reporting

The authors have taken time and put in effort to improve the m6A correlation and have added the dot blots. However, selection of the molecule is still not clear and most of the work is based on bioinformatics correlation. I think it’s too early for a complete story.

Experimental design

.

Validity of the findings

.

Reviewer 3 ·

Basic reporting

I agree with the point made by the other Reviewers and especially Review 1 that "The selection of COMMD10 and gastric cancer seems like a cherry-picking choice". I questioned the logical explanation of that point and asked questions 1-3 and 5. Those questions were not simply to ask for explanations of STAD and the function of COMMD10, but to ask you to reconsider whether it could be expressed more appropriately as the framework/story of the manuscript. In other words, why COMMD10 among the many prognostic genes in Figure 6D and why STAD among the many cancer types in Figure 1A?
I would like to see the above reconsideration, but since the rest of the questions were answered appropriately, I think a minor revision is appropriate.

Experimental design

no comment

Validity of the findings

no comment

---

## Round 0.3 · accepted · Accept

Per the reviewer and my analysis, the authors have successfully addressed all concerns related to the manuscript. Therefore, I recommend the following paper be officially accepted for publication.

·

Basic reporting

The authors return a complete and meticulously revised version of the manuscript.
They have addressed all the comments and provided all additional analysis.
I believe that the manuscript is ready for publication.
I would like to congratulate the authors for their efforts.

Experimental design

The authors included DEG analysis which crucial for the conclusions of the manuscript. Additionally the authors carefully address all the remaining comments.

Validity of the findings

The authors provide the R code, which after inspection looks reproducible and robust. They also answer technical questions about the analysis that they performed.

Reviewer 3 ·

Basic reporting

The authors addressed all of my comments and I recommend it to move to publication.

Experimental design

no comment

Validity of the findings

no comment